# Platelet Distribution Width Is Associated with P-Selectin Dependent Platelet Function: Results from the Moli-Family Cohort Study

**DOI:** 10.3390/cells10102737

**Published:** 2021-10-13

**Authors:** Benedetta Izzi, Alessandro Gialluisi, Francesco Gianfagna, Sabatino Orlandi, Amalia De Curtis, Sara Magnacca, Simona Costanzo, Augusto Di Castelnuovo, Maria Benedetta Donati, Giovanni de Gaetano, Marc F. Hoylaerts, Chiara Cerletti, Licia Iacoviello

**Affiliations:** 1Department of Epidemiology and Prevention, IRCCS NEUROMED, 86077 Pozzilli, Italy; alessandro.gialluisi@gmail.com (A.G.); sabatino.orlandi@moli-sani.org (S.O.); amalia.decurtis@moli-sani.org (A.D.C.); simona.costanzo@moli-sani.org (S.C.); mbdonati@moli-sani.org (M.B.D.); giovanni.degaetano@moli-sani.org (G.d.G.); chiara.cerletti@moli-sani.org (C.C.); licia.iacoviello@moli-sani.org (L.I.); 2Department of Medicine and Surgery, University of Insubria, 21100 Varese, Italy; francesco.gianfagna@uninsubria.it; 3Mediterranea Cardiocentro, 80133 Napoli, Italy; sara.magnacca@moli-sani.org (S.M.); dicastel@ngi.it (A.D.C.); 4Center for Molecular and Vascular Biology, Department of Cardiovascular Sciences, University of Leuven, 3000 Leuven, Belgium; marc.hoylaerts@kuleuven.be

**Keywords:** platelet distribution width, P-selectin, coagulation, VWF, mean platelet volume, thrombo-inflammation

## Abstract

Defined as an index of platelet size heterogeneity, the *platelet distribution width* (PDW) is still a poorly characterized marker of platelet function in (sub)clinical disease. We presently validated PDW as a marker of P-selectin dependent platelet activation in the Moli-family cohort. Platelet-bound P-selectin and platelet/leukocyte mixed aggregates were measured by flow cytometry in freshly collected venous blood, both before and after in vitro platelet activation, and coagulation time was assessed in unstimulated and LPS- or TNFα-stimulated whole blood. Closure Times (CT) were measured in a Platelet Function Analyzer (PFA)-100. Multivariable linear mixed effect regression models (with age, sex and platelet count as fixed and family structure as random effect) revealed PDW to be negatively associated with platelet P-selectin, platelet/leukocyte aggregates and von Willebrand factor (VWF), and positively with PFA-100 CT, and LPS- and TNF-α-stimulated coagulation times. With the exception of VWF, all relationships were sex-independent. In contrast, no association was found between mean platelet volume (MPV) and these variables. PDW seems a simple, useful marker of ex vivo and in vitro P-selectin dependent platelet activation. Investigations of larger cohorts will define the usefulness of PDW as a risk predictor of thrombo-inflammatory conditions where activated platelets play a contributing role.

## 1. Introduction

Platelet indices are extensively used as diagnostic and prognostic markers in several human pathological conditions. In particular, platelet count (Plt) and mean platelet volume (MPV) are assessed in clinical and population studies, often combined, as markers of platelet function [1]. In addition to the measurement of MPV, optical hematocytometry counters and automated measurements of impedance have allowed definition of the platelet distribution width (PDW), which is a volume parameter, measured as the difference between the highest and the lowest platelet volume, at 20% relative height in the platelet-size distribution histogram [2]. PDW is an index of platelet size heterogeneity, measuring the spread in platelet volumes [3,4,5]. It is also a marker of platelet anisocytosis, accompanying platelet activation, during which pseudopods form, increasing platelet diameter and apparent volume [5]. PDW is commonly used for the differential diagnosis of clinical conditions that present with increased platelet destruction or decreased platelet production. These conditions, characterized by modulated megakaryocyte differentiation and thrombopoiesis, in turn affecting platelet size and morphology, include (immune) thrombocytopenia [6] essential thrombocythemia [7] liver cirrhosis with altered platelet homeostasis [8] and idiopathic thrombocytopenic purpura [9].

We recently reported in the Moli-family study a significant association between both PDW and P-selectin dependent platelet activation and platelet endothelial aggregation receptor 1 (*PEAR1*) methylation [10]. PEAR-1 concentration in platelet membrane is a determinant of secondary platelet activation [11] and has been linked to several indices of platelet activation [12] this finding suggesting that PDW may be a simple, inexpensive, useful marker of thrombo-inflammatory platelet activation in population studies.

Besides its hematological link with deficient platelet production disorders, PDW has been identified as a marker of outcome or prognosis in clinically defined populations of different diseases where (thrombo)inflammation plays a role. These include cancer [13] cardiovascular disease [14,15] type 2 diabetes and metabolic syndrome [16] as well as neurological disorders [17], all conditions characterized by a (sub)clinical inflammation. The PDW relevance as a possible general marker of platelet activation remains, however, poorly defined. We presently investigated the platelet-dependent functional meaning of PDW variability in the Moli-family general population, clarifying its role in relation to in vivo platelet activation.

## 2. Materials and Methods

### 2.1. Study Population

The Moli-family cohort [10,18] includes 754 white subjects (≥15 years old) from 54 extended pedigrees (23 families with personal or familial history of early myocardial infarction (MI)—MI families—and 31 families without MI) recruited in Molise, a region in Southern Italy. All participants were relatives of index subjects enrolled in the general population-based Moli-sani study [19]. Enrollment and data collection were performed as previously described [20] and are further detailed in the Appendix A. The Moli-family study was approved by the Ethical Committee of the Catholic University in Rome, Italy. All subjects provided written informed consent; adult subjects also provided informed consent for their minor children.

### 2.2. Biochemical Measurements

All hematological analyses were performed by the same cell counter (Coulter HMX, Beckman Coulter, IL Milan, Italy), within 1 h from venipuncture. Soluble P-selectin and von Willebrand factor (VWF) were measured in stored plasma via the Human P-selectin Platinum Enzyme-linked Immunosorbent Assay (ELISA) kit (Affimetrix, eBioscience) [10,21] and the Hemosil^®^ von Willebrand Factor Antigen (ELISA kit) (Instrumentation Laboratory), respectively, according to the manufacturer’s instructions. High-sensitivity (hs) C reactive protein (CRP) was measured in serum as described [22].

### 2.3. Cell Functional Assays

Mixed platelet–leukocyte conjugates and platelet P-selectin expression were measured by flow cytometry before and after ADP/collagen stimulation of whole blood as described [18]. The PFA-100 CT was measured according to the gold standard guidelines using the collagen and ADP cartridge (CADP) [23]. Whole blood procoagulant activity was measured by the coagulation time, with or without the addition of bacterial endotoxin (LPS) (100 ng/mL) or tumor necrosis factor (TNF)-α (100 ng/mL) as described [21]. Further details on the protocols used are reported in the Appendix A.

### 2.4. Statistical Analyses

The final population used in the analyses included a total of 727 subjects (402 women). All analyses were performed using SAS/STAT software (Version 9.4 for Windows^©^2009, SAS Institute Inc., Cary, NC, USA). We first studied if PDW and the other platelet indices distribution was associated with age, sex and main lifestyle variables such as smoking, alcohol consumption, calory intake and body mass index (BMI), comparing their distribution tertiles by ANOVA or Chi-squared test, as appropriate. Second, to investigate apparent distribution relationships between PDW and the other platelet indices, we calculated a Pearson correlation matrix using the following variables: PDW, Plt and MPV. Third, we used multivariable linear mixed effects regression models to test associations between PDW and the above-mentioned platelet function/activation and blood coagulation tests. Age, sex and Plt were treated as fixed effects, and family stratification as a random effect. Because we observed a different distribution of PDW and of several of the variables studied in women vs. men (Appendix A), all the analyses were also repeated for women and men as two separated groups. In multivariable models, significance was set to α = 0.007, correcting for seven latent variables with Eigenvalue > 1, as identified through a principal component analysis (PCA) applied to the Pearson’s correlation matrix of all the outcome variables (N = 13). Nominally significant associations with *p* < 0.05 were also reported. All variables used in the analyses were standardized (see below).

## 3. Results

### 3.1. Population Characteristics, Correlations and Demographic Variables

Platelet indices and platelet activation/function markers presently studied, as well as differences between women and men are reported in Appendix A. All three platelet indices measured (PDW, MPV and Plt) followed a normal distribution (Appendix A) in line with previous observations in the Moli-sani cohort [24]. Women had lower PDW, higher Plt, ADP/collagen stimulated P-selectin and platelet/leukocyte aggregates compared to men (Appendix A).

A full descriptive of main population characteristics by PDW tertiles is reported in Table 1. Subjects in the highest tertile of PDW distribution were older, more frequently men and less frequently current smokers; they also reported more intense physical activity. However, BMI, daily energy intake, education and alcohol consumption did not vary significantly across PDW tertiles (Table 1).

We computed Pearson’s correlations between PDW and the other platelet indices Plt and MPV. PDW was significantly and negatively correlated with Plt (R = −0.199, *p* < 0.0001), but not with MPV (R = 0.032, *p* = 0.38) in the whole population and in both sex groups separately (Appendix A). An inverse correlation was also observed between MPV and Plt (R = −0.371, *p* < 0.0001; Appendix A).

### 3.2. PDW Is Associated with Platelet Activation and Blood Coagulation

We investigated how PDW distribution relates to a number of P-selectin dependent platelet activation tests as measured in the Moli-family cohort [18]. For this purpose, we used multivariate regression analysis with age, sex and Plt as fixed effects and family structure as random effect (Table 2). Plt was used as covariate in the model to verify its potential role as confounder in the associations, given its high correlation with PDW and its association with platelet function/activation and blood coagulation tests (Appendix A). PDW showed a strong negative association with basal platelet P-selectin expression and platelet/leukocyte aggregates in unstimulated conditions (with P-selectin: β(SE) = −0.140(0.037), *p* = 2 × 10^−4^; with platelet/monocyte aggregates: β(SE) = −0.147(0.036), *p* = 6 × 10^−5^; with platelet/polymorphonuclear cell (PMN) aggregates: β(SE) = −0.132(0.036), *p* = 2 × 10^−4^; Table 2). Concordant associations with smaller effect size were found between PDW and platelet/monocyte aggregates in ADP/collagen stimulated conditions (β(SE) = −0.090(0.038), *p* = 0.019, Table 2). We additionally tested the association between PDW and PFA-100 CT, observing a significant but milder association compared to the other platelet activation markers (β(SE) = 0.005(0.001), *p* = 1 × 10^−4^, Table 2). Its effect size increased after correction for VWF levels (β(SE) = −0.116(0.039), *p* = 0.003), in agreement with the predominant role of VWF in platelet function tests carried out in a shear force field [25].

We additionally investigated whether PDW distribution would show an association with whole blood coagulation times, recorded after recalcification of non-stimulated blood and of LPS or TNF-α stimulated blood, measured in the same cohort [21]. Higher PDW was associated with unstimulated (β(SE) = 0.358(0.200), *p* = 0.074) or LPS (β(SE) = 0.653(0.174), *p* = 2 × 10^−4^) or TNF-α stimulated coagulation time (β(SE) = 0.578(0.184), *p* = 0.002; Table 2). Yet, PDW was negatively associated with circulating VWF levels (β(SE) = −0.298(0.098), *p* = 0.002; Table 2), including when the association was adjusted by the hematocrit (Hct) (β(SE) = −0.100(0.039), *p* = 0.010).

No other significant association was found between MPV and other coagulation-dependent measurements (Appendix A). We did not find any significant association of PDW with both plasma P-selectin and CRP levels (Table 2).

The same associations were then tested with MPV since it showed no correlation with PDW and was then hypothesized to tag an independent effect. However, this parameter did not reveal any significant association with any of the variables mentioned above (Appendix A).

Models unadjusted by Plt are additionally reported in Appendix A (PDW) and Appendix A (MPV).

### 3.3. Sex-Specific Analysis

In view of the different distribution of PDW and of several of the variables studied in women vs. men (Appendix A), we performed the same multivariable regression analysis with age and Plt as fixed effects and family structure as random effect to investigate the association of PDW variability with platelet function/activation measurements in women and men separately. PDW was consistently inversely associated with platelet P-selectin and platelet/monocyte or PMN in unstimulated conditions in both women and men (Table 2). Moreover, the ADP/collagen stimulated platelet/leukocyte aggregates, which were raised in blood after platelet activation (Appendix A), were significantly higher in both women and men found in the lowest PDW tertile, compared to higher tertiles (Figure 1). We also observed a positive association of PFA-100 CT (β(SE) = 0.005(0.002), *p* = 0.001 in men and β(SE) = 0.005(0.002), *p* = 0.019, in women), which indicates a reduced platelet aggregation activity. In both these cases, adjusting the analysis by VWF levels increased the effect size of the association in women (β(SE) = 0.116(0.049), *p* = 0.019) and men (β(SE) = 0.140(0.064), *p* = 0.028).

LPS and TNF-α coagulation times were both associated with PDW in women and men (Table 2). The negative association between PDW and VWF previously detected in the whole population was mainly driven by the effect observed in women, which was four times higher than in men (β(SE) = −0.452(0.131), *p* = 0.001 and β(SE) = −0.116(0.149), *p* = 0.436, respectively; Table 2 and Figure 2). This association remained significant also by additionally adjusting by Hct (β(SE) = −0.139(0.052), *p* = 0.008).

No significant association was found for both PDW and MPV with plasma P-selectin or CRP in either women or men (Table 2), as observed in the whole population (see above).

## 4. Discussion

Through a functional analysis of platelet-related measurements previously performed in the Moli-family cohort, we show now and for the first time that PDW values reflect P-selectin dependent platelet function, both in women and in men. Although PDW has been considered as a sensitive marker of disease prognosis and diagnosis of several clinical conditions, the relations between PDW, platelet count and MPV in different populations have not been clearly defined [14,15,16,17]. Studies on these indices are scarce in general populations [24] and no prior investigation has linked PDW to platelet activation and function in vivo.

We found age to be an important determinant of both PDW and P-selectin dependent assays variability: older Moli-family individuals have higher PDW, as well as higher platelet P-selectin dependent activation [18]. Our data indicate that independently on age, individuals with lower PDW values show the highest activity, following platelet activation both in vivo and in vitro. Platelet P-selectin is a well-known marker of platelet activation and directly mediates vascular inflammation, facilitating leukocyte adhesion to the vessel wall through the formation of hetero-conjugates between platelets and PMNs or monocytes [26].

In accordance with the negative association between PDW and the P-selectin dependent assays, we also found that PDW is positively associated with PFA-100 closure time, a functional test specifically measuring high-shear dependent platelet activation [23]. Higher PDW marks longer PFA-100 closure times, reflecting a reduced platelet activation by ADP/Collagen.

Consistently, we found that low PDW was associated with higher circulating VWF levels, an association apparently and especially driven by the women in the Moli-family cohort (Table 2). Women have significantly higher ADP/collagen stimulated P-selectin expression and platelet/leukocyte mixed aggregates than men (Appendix A), reflecting the impact that higher platelet activation (low PDW) can have on circulating VWF levels. VWF promotes shear stress-regulated platelet aggregation [27] and platelet clearance [28] and has been recently described to play a role as mediator of thrombo-inflammation in a mouse model of ischemic stroke [29]. VWF is stored in endothelial cells and in platelet α-granules [30]. During degranulation caused by platelet activation, VWF is released together with other pro-coagulant molecules as well as P-selectin [31].

Our findings also confirm the role of platelet activation in whole blood coagulation since PDW was positively associated with the coagulation times measured in whole blood after stimulation with LPS or TNF-α. During platelet secretion, several procoagulant and anti-coagulant factors as well as polyphosphates supporting factor XII activation are released [32,33]. In addition, strong platelet activation also foster coagulation via membrane changes rendering factors such as platelet factor 3 (PF3) available for interacting with plasmatic coagulation factors [34]. Platelet P-selectin upregulates tissue factor expression/release by monocytes [35] and is able to bind PSGL-1 on TF-positive extracellular vesicles [36] such that thrombin generation can therefore be accelerated [37] and coagulation can be fostered. As judged from the corresponding β-values of the PDW associations with the coagulation assays, the impact of platelets is considerable, including when coagulation is not supported by those platelets that participate in aggregation, but is assured by ballooning platelets, with high exposure of phosphatidylserine [38].

PDW and MPV are both derived from the same platelet volume distribution histogram; theoretically, they measure size distribution and average platelet size, respectively. Our present analysis, however, shows that PDW and MPV are largely divergent in their relationship with specific platelet activation markers. MPV did not show any association with platelet function tests measured in the present study. Our data indicate that average platelet size variation per se is not a determinant of P-selectin expression during platelet activation. This interpretation of our findings is supported by two arguments. First, besides the absence of any correlation between PDW and MPV, their distribution histograms, in the whole population and in women and men separately, show some differences (Appendix A). MPV approached a gaussian distribution, while PDW distribution displayed a slight positive skewness (Appendix A). Second, men and women have comparable MPV, but women, with significantly lower PDW compared to men (Appendix A), showed significantly higher P-selectin dependent markers after ADP/collagen stimulation. All these data further confirm that the average size of resting platelets, as measured by MPV, bears no direct relation with the reactivity of platelets in P-selectin dependent assays. Yet, platelet size and number have long been considered to be one of the major determinants of platelet behavior in hemostasis, thrombosis and inflammation. Large or small platelets have been described to have distinct functional roles [39,40,41,42,43,44,45,46,47,48,49], suggesting that platelet size heterogeneity, measured by PDW, may influence individual platelet response in hemostasis and beyond.

Limitations should be taken into account when considering our findings. First, both MPV and PDW are partially dependent on the technologies used to record them [50]. Based on that, it is not possible to define, at this stage and within this study framework, any cut-off or variable range that can be exported and applied to other cohorts. Second, we need to acknowledge that PDW (and MPV) do not measure exact platelet volumes but rather apparent volume changes that could be influenced by several pathophysiological processes including morphological changes caused by platelet activation, variability in platelet formation and elimination. Third, because of the design of the study, it is not possible to define the potential clinical value of PDW measurements. Cohort studies do not allow drawing conclusions at an individual level, that is required in the clinical laboratory. Our study represents, on the contrary, a first attempt to clarify the possible platelet-related functional meaning of this index based on an observational approach of a general population. Lastly, blood type information relative to the Moli-family participants was not available, we therefore were not able to rule out the influence of this parameter on the association between PDW and VWF.

## 5. Conclusions

In conclusion, our data link for the first time PDW to P-selectin dependent platelet activation and function in a general population, in the absence of acute cardiovascular disease events. This relationship indicates that PDW is a simple, inexpensive, useful marker of platelet activation in both women and men and is influenced by age. Further investigations of larger cohorts will define whether PDW measurement will be useful in population stratification for risk prediction in diseases where platelets play a contributing role. Larger prospective cohorts with an adequate number of cases and platelet P-selectin dependent variables will allow to determine whether MPV and PDW have additive predictive value as risk factors of thrombo-inflammatory disease.

## Figures and Tables

**Figure 1 cells-10-02737-f001:**
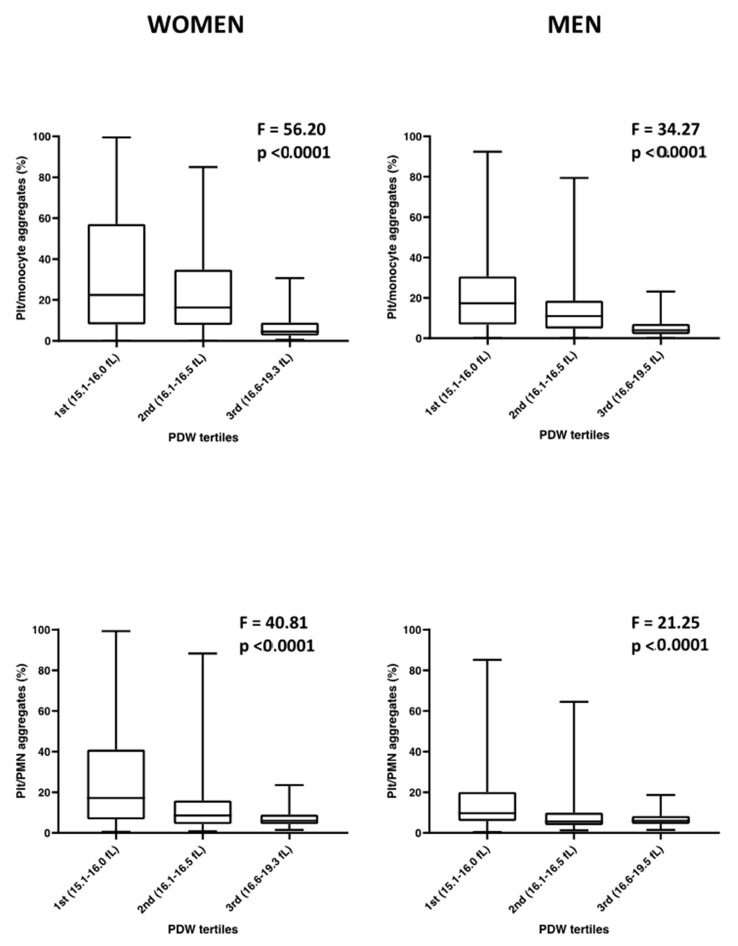
Sex-specific platelet/leukocyte mixed aggregate formation after ADP/collagen stimulation of whole blood, subdivided according to platelet PDW tertiles. Distribution of platelet/monocyte and platelet/PMN mixed aggregates in women and men of the Moli-family cohort over PDW tertiles, as defined by the ranges indicated in parenthesis on the X axis. One-way ANOVA test significance is reported for each graph. N (Women) = 402; N (Men) = 325.

**Figure 2 cells-10-02737-f002:**
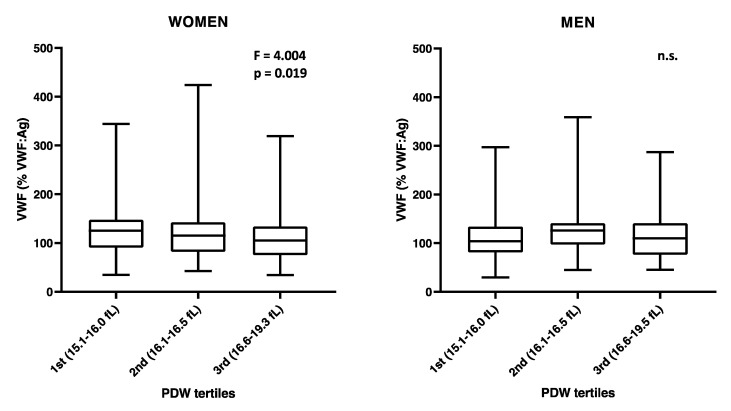
Sex-specific VWF levels by PDW tertiles. Distribution of VWF levels in women and men of the Moli-family cohort over PDW tertiles, as defined by the ranges indicated in parenthesis on the X axis. One-way ANOVA test significance is reported for each graph. N (WOMEN) = 402; N (MEN) = 325.

**Table 1 cells-10-02737-t001:** Moli-family cohort characteristics by PDW tertiles.

	PDW 1st (15.1–16.0 fL)			PDW 2nd (16.1–16.5 fL)			PDW 3rd (16.6–19.5 fL)			
**Continuous variables**	**N**	**Mean**	**SD**	**N**	**Mean**	**SD**	**N**	**Mean**	**SD**	**Pr > F**
Age (years)	221	39.94	17.63	268	41.71	19.17	238	46.02	18.73	0.001
BMI (kg/m^2^)	219	26.11	4.74	265	26.41	5.82	238	26.77	5.18	0.405
Energy intake (Kcal/d)	203	1655.19	561.05	239	1728.83	526.26	213	1671.63	587.51	0.339
**Categorical variables**		**N**	**%**		**N**	**%**		**N**	**%**	
Males		77	34.84		125	46.64		123	51.68	0.001
Smoker status (smokers)										0.018
Ever		123	55.66		146	54.48		129	54.2	
Current		73	33.03		82	30.6		57	23.95	
Former		25	11.31		40	14.93		52	21.85	
Physical activity										0.029
Sedentary		76	35.02		92	35.11		62	26.61	
Light physical activity		45	20.47		59	22.52		71	30.47	
Moderate physical activity		72	33.18		65	24.81		63	27.04	
Intense physical activity		24	11.06		46	17.56		36	15.45	
Alcohol consumption										0.092
0 g/day (Former/ever drinker)		78	37.5		82	33.2		64	29.49	
≤12 g/day		78	37.5		107	43.32		79	36.41	
12.1–24 g/day		32	15.38		28	11.34		42	19.35	
>24 g/day		20	9.62		30	12.15		32	14.75	

**Table 2 cells-10-02737-t002:** PDW association with thrombo-inflammatory and coagulation variables.

Variable	PDW														
	ALL @					WOMEN #					MEN #				
	*β*	*SE*	*Probt*	*Lower*	*Upper*	*β*	*SE*	*Probt*	*Lower*	*Upper*	*β*	*SE*	*Probt*	*Lower*	*Upper*
Platelet P-selectin basal	−0.140	0.037	0.00018 *	−0.213	−0.067	−0.127	0.048	*0.009*	−0.221	−0.032	−0.156	0.059	*0.008*	−0.271	−0.041
Platelet P-selectin ADP/Collagen	0.002	0.038	0.960	−0.074	0.078	−0.017	0.052	0.748	−0.120	0.086	0.033	0.056	0.562	−0.078	0.143
Platelet/monocyte aggregates basal	−0.147	0.036	0.00006 *	−0.219	−0.076	−0.152	0.049	0.002 *	−0.248	−0.057	−0.134	0.055	*0.016*	−0.242	−0.026
Platelet/monocyte aggregates ADP/Collagen	−0.090	0.038	*0.019*	−0.165	−0.015	−0.132	0.046	0.005 *	−0.223	−0.041	0.007	0.068	0.917	−0.127	0.142
Platelet/PMN aggregates basal	−0.132	0.036	0.00024 *	−0.202	−0.062	−0.117	0.046	*0.011*	−0.206	−0.027	−0.175	0.060	0.004 *	−0.293	−0.057
Platelet/PMN aggregates ADP/Collagen	−0.051	0.037	0.169	−0.124	0.022	−0.106	0.045	*0.019*	−0.194	−0.017	0.062	0.068	0.362	−0.071	0.195
PFA-100 CT	0.005	0.001	0.00012 *	0.002	0.007	0.005	0.002	0.001 *	0.002	0.009	0.005	0.002	*0.019*	0.001	0.009
Coagulation time	0.358	0.200	0.074	−0.034	0.751	0.217	0.265	0.414	−0.305	0.739	0.715	0.293	*0.016*	0.137	1.293
LPS stimulated coagulation time	0.653	0.174	0.00018 *	0.312	0.995	0.607	0.238	*0.011*	0.139	1.075	0.916	0.250	0.00031 *	0.423	1.410
TNF-α stimulated coagulation time	0.578	0.184	0.002 *	0.217	0.940	0.573	0.240	*0.018*	0.100	1.045	0.720	0.285	*0.012*	0.158	1.283
VWF	−0.298	0.098	0.002 *	−0.490	−0.106	−0.452	0.131	0.001 *	−0.710	−0.193	−0.116	0.149	0.436	−0.409	0.177
Soluble P-selectin	−0.093	0.050	0.064	−0.190	0.005	−0.082	0.073	0.258	−0.226	0.061	−0.129	0.066	0.053	−0.260	0.001
CRP	−0.004	0.037	0.910	−0.078	0.069	−0.077	0.052	0.141	−0.180	0.026	0.087	0.055	0.113	−0.021	0.195

@ Model adjusted by age, sex and Plt as fixed effect, family structure as random effect. # Model adjusted by age and Plt as fixed effect, family structure as random effect. Significant *p* values (<0.007) are highlighted by an asterisk, nominally significant *p* values (<0.05) are reported in italics.

## Data Availability

The data underlying this article will be shared on reasonable request to the corresponding author. The data are stored in an institutional repository (https://repository.neuromed.it) and access is restricted by the ethical approvals and the legislation of the European Union.

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
