# Peer review of "Platelet Distribution Width Is Associated with P-Selectin Dependent Platelet Function: Results from the Moli-Family Cohort Study"

_cells, 2021, doi:10.3390/cells10102737_

Round 1

Reviewer 1 Report

This manuscript contain interesting and important data, however my opinion is not changed. The manuscript is not suitable for the Journal Cells. Cells mostly publishing papers connected with the mechanisms of intracellular signaling events and descriptions of new aspects of cells structure etc. Presented manuscript is more clinically oriented and I can recommend to authors to choose another journal.

Reviewer 2 Report

No further comments

Reviewer 3 Report

In this paper by B. Izzi et al. the authors clearly answered the questions provided by the reviewers.

The title has been changed as suggested and is correct.

The tables have been arranged and significant results have been highlighted.

The discussion has been arranged in a simpler and more direct way that is straightforward for those who read the work.

This manuscript is a resubmission of an earlier submission. The following is a list of the peer review reports and author responses from that submission.

Round 1

Reviewer 1 Report

This manuscript might be interested for clinicians and I can suggest to recommend to authors to send it to some clinically oriented journals. 

In this manuscript Izzi et al presented validation of the platelet distribution width (PDW) as a marker of P-selectin dependent platelet activation in the Moli-family cohort. Mean platelet volume (MPV) and PDW are already used in many clinical studies and trials as markers of platelet activation. In the introduction, discussion and conclusions of the presented manuscript not clearly described what real new information for will give this study.

The title of manuscript and supplementary material should be the same.

Reviewer 2 Report

The present work by Izzi and co-workers represents a cohort study to unravel the suitability of PDW as marker for platelet function in thrombo-inflammatory diseases. While they found a negative correlation between PDW and CD42P, no fundamental experimental data are presented to corroborate their conclusion or to investigate the underlying mechanism. The manuscript is written well and the analyses are done with meticulous care. Although the manuscript fits the scope of the special issue of Cells, it does not add substantial new information in the present form, since it is established that platelet size heterogeneity affects platelet function and activity. To strengthen the manuscript further experiments have to be performed.

Major points:

  1. As the authors correctly state the pro-coagulant state of platelets pivotally contributes to coagulation and thrombus formation. Thus, to strengthen the manuscript, the authors should measure the thrombin generation of platelets in accordance to the PDW in their cohort or at least determine the PS-exposure by flow cytometry.
  2. According to the presented data showing a negative correlation between PDW and CD42P expression it seems plausible that individuals with higher MI prevalence in the cohort show a smaller PDW when compared with individuals from families without MI. Although the authors mention that there is no association, no data sets are presented. So the authors have to show and discuss the mentioned results. They also have to explain and discuss this discrepancy.
  3. Could the authors explain why older individuals show higher PDW values and CD42P-dependent platelet activation, while the present study unravels that individuals with high PDW show decreased CD42P expression? Is it only a confounding effect of age or does other factors also play a role?
  4. Please state the correct sample size in figure 1 and 2, table 2 as well as in all the supplementary tables and be aware of total sample size (N) and subgroups (n). The analyses were performed with the whole population or only part of the cohort?
  5. How are other pro-inflammatory factors from alpha-granules such as IL1-ß, CXCL12, etc. are regulated in association with the PDW in the used cohort?
  6. The discussion is way too long for the presented data sets and should be shortened and specified.

Minor points:

  1. It would help the reader if the figures and tables would be presented in a more consistent and clear way. Moreover, it would be nice to present the figures in the correct order as mentioned in the manuscript. At some points the order of the figures and tables is a little bit confusing. A clear specification of the significances in figure 1 and 2 would also help the reader to understand the results.
  2. Can you please state the gating strategy for the flow cytometric measurements performed in the present study?
  3. Please indicate the significance in the supplementary tables in a different way using e.g. stars.
  4. Page 10, line 15: What does the authors mean by "...longer closure times and or pro-coagulant activity..."? Do you want to state impaired/decreased pro-coagulant activity? Please clarify. Otherwise it seems that pro-coagulant activity is associated with impaired platelet function.
  5. 5. Page 10, line 6: It sounds a bit odd to start a sentence with "Or". Please change.

Reviewer 3 Report

The work by Izzi et al. investigates if platelet distribution width is associated with several platelet activation markers and clinical traits. The authors used the already published Moli-family cohort composed of 754 subjects from several families with or without myocardial infarction history. The authors conclude that despite PDW is not associated with MI occurrence, it is associated with platelet activation markers such as P-selectin exposure or platelet/leukocyte aggregates. The study is of interest because it adds additional relevance to PDW that is a simple parameter to analyze. In addition, its specificity vs MPV that does not clearly associate with these same markers also increases the interest.

In general the study is easy to follow, although some results like the associations of PDW with education or physical activity are not clear and dilute the message. The discussion is clear with important messages and a limitation paragraph that is important for this work. In particular, given that PDW associates with age, gender and platelet activation but does not associate with the occurrence of MI, substracting clinical interest to the study.

I have some comments that follow:

  1. The dose response curves of LPS and TNFa are of interest, please include them in the supplementary material document.
  2. Please indicate the units (fL) in the X axis in suppl. Figure 1 and in Suppl. Table 4
  3. In the Material section, I suggest to include subheadings for each specific assay, including a final subheading of “Statistical Analysis”
  4. Please specify on how the time of coagulation is assessed. It is not clear how this was performed (in particular how the time in seconds is measured). I was checking the referenced bibliography but this is not clear.
  5. The sentence “These analyses were also repeated with MPV, since it showed no cor-relation with PDW and was then hypothesized to tag an independent effect (see below). Models unadjusted by Plt are additionally reported in the Supplementary table 1 (PDW) and Supplementary table 2 (MPV).” Should rather be included in the Results Section.
  6. Supplementary Table 2 should be ordered for example by putting together the different percentiles.
  7. In the first subheading of the Results section, the authors state that “All four platelet indices followed or approximated a normal distribution”. What does that mean? Please insert the results from the normality tests used. Additionally, why the authors talk of four indices if the data only refer to two indices: PDW and MPV? Please rephrase.
  8. I do not find the title appropriate. The main focus of this paper is to investigate the association between PDW and other clinical features such as gender, or activation markers.

Reviewer 4 Report

Comments to the Authors

In this paper by B. Izzi et al. the authors investigated the function of PDW, a parameter of platelet volume, as a marker of ex vivo and in vitro P-selectin platelet depended activation. Platelet P-selectin plays an important role in the adhesion of leukocytes to the vessel wall as a result of the interaction between platelet and PMN and monocytes. The study was conducted in a large cohort the Moli-family cohort. In this study were analysed the interaction between Plts and PMN and monocytes. Moreover, their analysed the PDW with other coagulation markers such as PF-100 CT, soluble P-selectin and VWF.  In conclusion this study demonstrated a link between PDW and P-selectin depended platelet activation. The results regards a healthy populations and a population with myocardial infarction.

Here are many specific comments on manuscript.

Title: The paper focus in the role of PDW as a platelet size index   and its correlation with P-selectin, a marker of platelet activation  
I suggest changing the title to a more appropriate title for the study such as: Platelet distribution width a marker of P-selectin depended platelet function: results from the Moli-family cohort study.

Introduction:
In the paragraph “Platelet P-selectin is the major ………and PMNs, respectively monocytes”. Please write the meaning of the abbreviated term PMNs.

Results:
- In the material and methods in the paragraph “Second, because of the study structure, we also compared PDW values between subjects with vs without a family history of MI, through a Wilcoxon signed rank test”. 
The results related the MI group subjects were not descripted in the results and in the discussion paragraph. 

- In the paragraph: “All four platelet indices followed or approximated a normal distribution……………………in line with previous observations in the Moli-sani cohort.”
Most of the results descripted in this paper were found in previous papers by the same authors, even if the number of subjects was smaller. Please insert the information of previous results.

- Table 1: “Moli-family cohort characteristics by PDW tertiles” were descripted only the males population what about the other categories of population? 
I suggest to deleting the table and briefly descripting the results.

Discussion:

- The first line “Through a functional analysis of platelet-related measurements in the Moli-family cohort …………………………..both in women and in men”. In material and methods it is said that the study involved families with healthy subjects and families that have a history of MI but  it is not clearly descripted that the analysis of the results were between man and women groups. Please comment this part.

- Most of the data included in this paper it is not necessary and it is not of relevant importance like the  table 1 “Moli-family cohort characteristics by PDW tertiles”. People's education  is not important  because the PDW does not change with the  education.  Please review the data in table 1.

General consideration:
Did you analysed by flow cytometry the different sizes of platelets and their changes in the expression of P-selectin and of the other coagulation markers or of platelets/leukocyte aggregates before and after stimulation with agonist?  

Supplementary material
Study population
There is no information regarding the number of people with MI and it is not clear how these patients are considered in this study as a group of people at risk of cardiovascular diseases?
In the paragraph “The index subject of each family………………… with CVD before 65 years” please write the meaning of the abbreviated term CDV.
Flow cytometry measurements
Although the method was described in the reference article 1, it is not clear which different populations were considered for this study. Briefly describe better which populations are were considered and how.  Which specific markers and population of double positivity  for each aggregate for  platelets, leukocytes, PMN and monocytes populations were considered?

Whole blood functional tests: PFA-100 Closure Time (CT) and whole blood procoagulant activity
- In the paragraph it is not clear from the different references (3, 7) what kind of instrumen is used to measure the coagulation time in whole blood. Please insert this information and insert the normal va-

- The paper focus in the role of PWD and not to MPV. I suggest to changing the tables and to descript only the results regarding PDW. Futhermore, it is important to underline the association or not of PDW with MPV.

-In the table 7 we observed the results of MPV which is not the subject  of this paper and could be eliminate and only comment the results  at the discussion paragraph (or in the results). Furthermore, there is not have any information regarding PDW results with adjusted by platelets number.

- All the tables descripted the results between three groups “All, Women and Men” but in material and methods a group of subjects with MI were descripted, there is not any results regarding?